# Tunable thermal expansion in framework materials through redox intercalation

Jun Chen[1], Qilong Gao[1], Andrea Sanson[2], Xingxing Jiang[3], Qingzhen Huang[4], Alberto Carnera[2], Clara Guglieri Rodriguez[5], Luca Olivi[5], Lei Wang[6], Lei Hu[1], Kun Lin[1], Yang Ren[7], Zheshuai Lin[3], Cong Wang[6], Lin Gu[8], Jinxia Deng[1], J. Paul Attfield[9] & Xianran Xing[1]

Thermal expansion properties of solids are of fundamental interest and control of thermal expansion is important for practical applications but can be difficult to achieve. Many framework-type materials show negative thermal expansion when internal cages are empty but positive thermal expansion when additional atoms or molecules fill internal voids present. Here we show that redox intercalation offers an effective method to control thermal expansion from positive to zero to negative by insertion of Li ions into the simple negative thermal expansion framework material $ScF_3$, doped with 10% Fe to enable reduction. The small concentration of intercalated Li ions has a strong influence through steric hindrance of transverse fluoride ion vibrations, which directly controls the thermal expansion. Redox intercalation of guest ions is thus likely to be a general and effective method for controlling thermal expansion in the many known framework materials with phonon-driven negative thermal expansion.

[1] Department of Physical Chemistry, University of Science and Technology Beijing, Beijing 100083, China. [2] Department of Physics and Astronomy, University of Padova, I-35131 Padova, Italy. [3] Center for Crystal R&D, Key Lab of Functional Crystals and Laser Technology of Chinese Academy of Sciences, Technical Institute of Physics and Chemistry, Chinese Academy of Sciences, Beijing 100190, China. [4] NIST Center for Neutron Research, National Institute of Standards and Technology, Gaithersburg, Maryland 20899-6102, USA. [5] Elettra Sicrotrone Trieste, Strada Statale 14 - km, in AREA Science Park, 34149 Basovizza, Italy. [6] Center for Condensed Matter and Materials Physics, Department of Physics, Beihang University, Beijing 100191, China. [7] Argonne National Laboratory, X-Ray Science Division, Argonne, Illinois 60439, USA. [8] Beijing National Laboratory for Condensed Matter Physics, Institute of Physics, Chinese Academy of Sciences, Beijing 100190, China. [9] Centre for Science at Extreme Conditions and School of Chemistry, University of Edinburgh, Peter Guthrie Tait Road, King's Buildings, Edinburgh EH9 3FD, UK. Correspondence and requests for materials should be addressed to J.P.A. (email: j.p.attfield@ed.ac.uk) or to X.X. (email: xing@ustb.edu.cn).

Most materials exhibit positive thermal expansion (PTE) due to the inherent anharmonicity of bond vibrations[1–4] that leads to expansion of average bond distances with increasing temperature. This is a critical issue in many high precision applications subject to large temperature fluctuations such as optical instruments, electronic devices, and spaceflight engineering[2–4]. Thermal expansion control engineering typically makes use of the unconventional property of negative thermal expansion (NTE) which is found in a variety of materials such as oxides[1–7], alloys[8,9], nitrides[3,10], organic compounds[11,12], ReO₃-based compounds[13–17], metal organic frameworks (MOFs)[18,19] and cyanides[20]. Composites of NTE and PTE materials are often used but may fail after repeated cycling, so direct control of thermal expansion within a single homogenous phase is desirable.

NTE can arise from electronic or magnetic mechanisms, and by a transverse phonon mechanism in insulating framework solids[4]. These usually have an open structure of corner-sharing metal-anion tetrahedra or octahedra[2], for example, the archetype material $ZrW_2O_8$ where large NTE was discovered in 1996 (refs 1,2). NTE has subsequently been reported in many open framework materials such as $ReO_3$-type fluorides[21–25], MOFs[26,27], cyanides[28–30], zeolites and $AlPO_4$ frameworks[31,32]. Although the basic mechanism of NTE in open framework materials through the presence of low energy transverse vibrations is widely accepted, this does not lead to straightforward control of thermal expansion. Chemical substitutions of the framework often have small effects on the lattice dynamics related to expansion, for example, the linear coefficient of thermal expansion (CTE, $\alpha_l$) for $Zr_{1-x}M_xW_2O_{8-y}$ ($M=$ Sc, In, Y) materials varies only over a short range $(-7.3$ to $-8.7 \times 10^{-6} K^{-1})$[33]. However, introduction of small molecules such as water into large pore materials is known to have dramatic effects, for example, water adsorption in $ZnPt(CN)_6 \cdot xH_2O$ cyanide[34] and $ZrW_2O_8$ (ref. 35) switches NTE to PTE behaviour, and thermal expansion is very different between dehydrated[36,37] and hydrated forms of cation-exchanged zeolite LTA[38].

Hence, a potentially general method for varying thermal expansion is to use intercalation chemistry where small cations such as $Li^+$ can be inserted or removed even from relatively dense frameworks, as much applied in Li-battery chemistry. The intercalated cations are expected to sterically hinder or reduce the transverse vibrations responsible for NTE. We have tested this approach using $ScF_3$ which has a very simple cubic $ReO_3$-type structure, and the results shown here demonstrate that thermal expansion is very effectively tuned from negative to zero or positive values through small changes to the degree of Li ion intercalation.

## Results

### Composition design for control of thermal expansion in $ScF_3$.
$ScF_3$ has a simple cubic $ReO_3$ crystal structure consisting of a corner-shared $ScF_6$ octahedra (Fig. 1a), equivalent to $ABX_3$ perovskite where the A-site is vacant. $ScF_3$ shows isotropic NTE over a wide range of temperature (10–1,100 K)[22,25], and lattice dynamics studies have explored in detail how enhanced transverse thermal vibrations of fluoride ions with increasing temperature lead to shrinkage of Sc–F–Sc linkages and hence NTE[25,39] (Fig. 1b). We propose that if those vibrations of fluorine ions can be reduced or hindered by the intercalation of Li ions into the cages of $ScF_3$ (Fig. 1a,c), then control of thermal expansion can be realized.

Insertion of Li ions into $ScF_3$ can be achieved by reductive lithiation, which is commonly used for Li-ion battery materials. $Sc^{3+}$ is not easily reducible, and direct reactions using

$n$-butyllithium failed to form $Li_xScF_3$ products, so partial substitution of Sc by a reducible metal is needed. The solid solution $(Sc_{0.9}Fe_{0.1})F_3$ (SFF) was thus synthesized and then lithiated, as described in Methods. The ionic radii of $Sc^{3+}$, $Fe^{3+}$ and $Fe^{2+}$ are, respectively, 0.745, 0.645 and 0.780 Å, so reduction of $Fe^{3+}$ to $Fe^{2+}$ in this framework is favoured by lowering of lattice microstrain. Characterization results for SFF and the lithiated product $Li_x(Sc_{0.9}Fe_{0.1})F_3$ are shown in Fig. 2a.

**Composition determination**. The lattice constant increases slightly on Li intercalation, from 3.99368(5) to 3.99927(4) Å for SFF and $Li_x(Sc_{0.9}Fe_{0.1})F_3$, respectively (Fig. 2a), in keeping with lattice expansion due to reduction of $Fe^{3+}$ to $Fe^{2+}$. The structure and composition of the lithiated $Li_x(Sc_{0.9}Fe_{0.1})F_3$ product have been determined by joint studies of structure refinement based on neutron powder diffraction (NPD) data, spherical aberration-corrected scanning transmission electron microscopy (STEM) and X-ray absorption near-edge structure (XANES). A neutron scattering Fourier difference map clearly demonstrates that the negative peak at the (0,0,0) is the position of Li ions (Fig. 2b; Supplementary Fig. 1), since Li has a negative neutron scattering length[40]. Indeed, the refinement is greatly improved by assuming that Li ions are located at the (0,0,0) position in the centre of the perovskite cage (Supplementary Table 1). The chemical composition determined by NPD refinement of the Li occupancy is $Li_{0.06}(Sc_{0.9}Fe_{0.1})F_3$ (sample LSFF-1; Supplementary Fig. 2), which was further supported by ICP analysis (Supplementary Note 1), so the Li content is below the theoretical maximum of $x=0.1$. The annular-bright-field (ABF) electron micrographs of LSFF-1 and SFF directly reveal the Li sites within the structural model (Fig. 2c–e), and hence also provide direct evidence for the A-site occupancy of Li ions.

XANES spectra were collected from samples SFF and LSFF-1. There is no change in the Sc K-edge XANES (Supplementary Fig. 3a), which shows that the chemical valence of Sc remains constant during the lithiation. On the other hand, the Fe K-edge is clearly shifted to lower energies, which indicates a partial reduction from $Fe^{3+}$ to $Fe^{2+}$ after the lithiation reaction (Supplementary Fig. 3b). The pre-edge peak (Fig. 2f–h), corresponding to the $1s \rightarrow 3d$ transition, can be used to estimate the $Fe^{3+}/\sum Fe$ ratio (Supplementary Note 2). For SFF, the $Fe^{3+}/\sum Fe$ ratio is near to unity showing that Fe ions are in the $+3$ state. However, for LSFF-1, the $Fe^{3+}/\sum Fe$ ratio is 0.36, consistent with the 40% residual proportion of $Fe^{3+}$ ions predicted for the composition $Li_{0.06}(Sc_{0.9}Fe_{0.1})F_3$ found by NPD refinement.

**Tunable thermal expansion via Li intercation**. The Lattice parameter measurements demonstrate that Li intercalation has a strong influence on thermal expansion (Fig. 3a). $ScF_3$ and SFF have NTE behaviour with average linear thermal expansions (in the range 150–425 K) of $\alpha_l = -7.47$ and $-5.01 \times 10^{-6} K^{-1}$, respectively. However, introduction of a small amount of Li at the A-sites results in a change to PTE with $\alpha_l = 1.03 \times 10^{-6} K^{-1}$ for LSFF-1 as shown in Fig. 3a. Samples with lower lithium contents were generated by heating LSFF-1 at temperatures above 425 K in inert an $N_2$ atmosphere resulting in loss of Li as LiF and Fe as $Fe_3O_4$, and $\alpha$-$Fe_2O_3$ (Supplementary Figs 4–7 and Table 2), and three further $Li_x(Sc_{1-y}Fe_y)F_3$ products (samples LSFF-2 to 4) were generated by this route (Fig. 3b). This was used as a convenient way to deintercalate lithium and demonstrate resulting changes in thermal expansion, although it is not a practical method for applications.

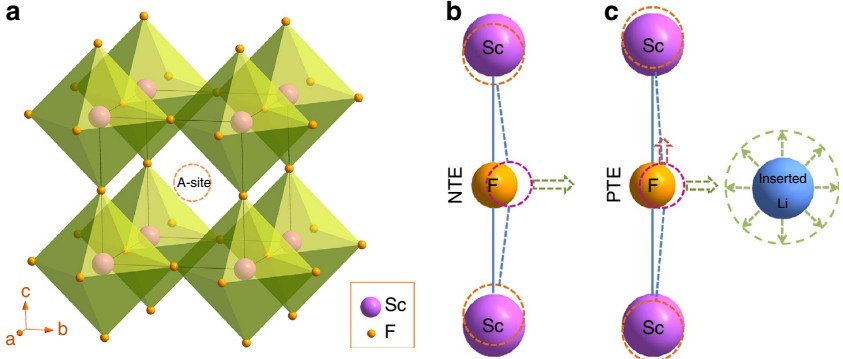

**Figure 1 | The effect of Li ion interaction on the tunable thermal expansion of $ScF_3$.** (**a**) The cubic structure of $ScF_3$ with open framework (space group: $Pm\bar{3}m$). The cage consisting of corner-shared $ScF_3$ regular octahedra is marked with the dash line circle (A-site). The guest ions or molecules can be inserted at the A-site cage. (**b**) The negative thermal expansion of $ScF_3$ induced by the transverse vibration of fluorine normal to the linkage of Sc–F–Sc. (**c**) The steric hindrance role of inserted ions, eg, $Li^+$, in the vibration of fluorine. The longitudinal vibration of fluorine results in the positive thermal expansion.

Phase contents were determined by structure refinement based on the NPD data, and the results are tabulated in Supplementary Table 3. Their thermal expansion curves in Fig. 3a demonstrate that $\alpha_l$ changes smoothly with Li content. Near zero thermal expansion (ZTE) with $\alpha_l = -0.75 \times 10^{-6}$ $K^{-1}$ is achieved in $Li_{0.04}(Sc_{0.94}Fe_{0.06})F_3$ (LSFF-2) annealed at 475 K, while $Li_{0.02}(Sc_{0.97}Fe_{0.03})F_3$ (LSFF-3) has moderate NTE of $\alpha_l = -2.59 \times 10^{-6} K^{-1}$. Annealing at 575 K gives sample LSFF-4 with $\alpha_l = -7.40 \times 10^{-6} K^{-1}$ the same as for $ScF_3$ (Supplementary Fig. 8), showing that all Li and Fe have been driven from the framework (Supplementary Note 3). Hence, chemical control of thermal expansion is achieved by adjusting the intercalated Li content. Maximum NTE is obtained for the composition with zero Li content, while weaker NTE, ZTE or PTE can be achieved as Li content increases.

**The mechanism of controllable thermal expansion.** A previous inelastic neutron scattering study revealed that the NTE behaviour of $ScF_3$ mainly originates from the transverse vibration of fluoride ions at low frequencies (0–30 meV)[39]. To investigate how Li ion intercalation tunes the thermal expansion of $ScF_3$, we have performed vibrational analysis of the fluorine atoms using first-principles calculations. The results reveal that the vibrations of fluorine ions are strongly perturbed by the inserted Li ions, and the directions of all transverse modes are inclined as shown for the representative mode with the lowest frequency in Fig. 4a,b. Figure 4a shows the lowest frequency ($37 cm^{-1}$) vibrational mode for undoped $ScF_3$ where the vibrational motion of fluorine ions is perpendicular to the Sc–F–Sc linkages leading to NTE in $ScF_3$. Interestingly, intercalation of Li ions into the $ScF_3$ cages strongly perturbs this vibrational mode (Fig. 4b). The vectors of thermal vibration for the closest fluorine ions change from being perpendicular to the Sc–F–Sc linkage to an angle of $\sim 50°$ and hence have significant transverse and longitudinal components. This demonstrates that intercalation of Li ions redistributes the fluorine vibrational motion locally, contributing to bond stretching thermal expansion and PTE. Hence even a small concentration of Li ($x \sim 0.04$) is sufficient to suppress the overall NTE behaviour, owing to the dual effects on the amplitude and direction of transverse vibrations of fluorine ions.

The effects of Li intercalation on thermal expansion are further supported by the anisotropic atomic amplitude of fluorine ions calculated in the structure refinements from NPD data.

Supplementary Table 4 and Fig. 4c show the values of anisotropic atomic displacement parameters of fluorine ions and CTEs of the LSFF compositions. With increasing content of Li the transverse thermal vibration amplitude ($U_{33}$) of F ions is weakened, while the longitudinal one ($U_{11}$) is enhanced. As shown in Fig. 4c, there is a good correlation between CTE and the ratio $U_{33}/U_{11}$. Larger $U_{33}/U_{11}$ corresponds to stronger transverse thermal vibration of fluorine ions, and a more negative expansion coefficient. The change of thermal expansion from PTE to NTE is accompanied by an increase in the ratio of $U_{33}/U_{11}$.

## Discussion

The above results demonstrate that introducing a small concentration of Li ions into $(Sc_{0.9}Fe_{0.1})F_3$ switches the thermal expansion behaviour from NTE to PTE, consistent with the Li guest ions in $Li_{0.06}(Sc_{0.9}Fe_{0.1})F_3$ providing steric hindrance to the transverse vibrations of fluorine ions. Thus, it is likely that NTE can be controlled in many frameworks by adjusting the concentration of guest ions, as previously found for several frameworks by varying the concentration of guest water molecules[34–38]. This method should be applicable to NTE frameworks containing reducible cations, otherwise substitution can be used to introduce such species such as the small amount of iron replacing scandium in the present example.

The present study may provide an effective method to achieve thermal expansion control engineering by means of the redox intercalation of guest ions or molecules into the pores of a NTE framework. Although we have used chemical intercalation to introduce lithium into $(Sc_{0.9}Fe_{0.1})F_3$ and thermal decomposition to delithiate leading to impurity formation, standard electrochemical methods used in batteries or sensors could be used to introduce or remove cations reversibly. A similar Li insertion and extraction has been reported to occur reversibly and rapidly at room temperature for $ReO_3$-type $FeF_3$ via an electrochemical method[41], validating the feasibility of this approach. Electrochemical methods could lead to smart devices that respond to external stimuli by varying their coefficient of thermal expansion to control complex devices such as precise multicomponent optics. Redox intercalation thus offers a general method for thermal expansion control engineering at the materials and device levels.

In summary, the present study demonstrates that redox intercalation of guest cations into the empty pores of a framework

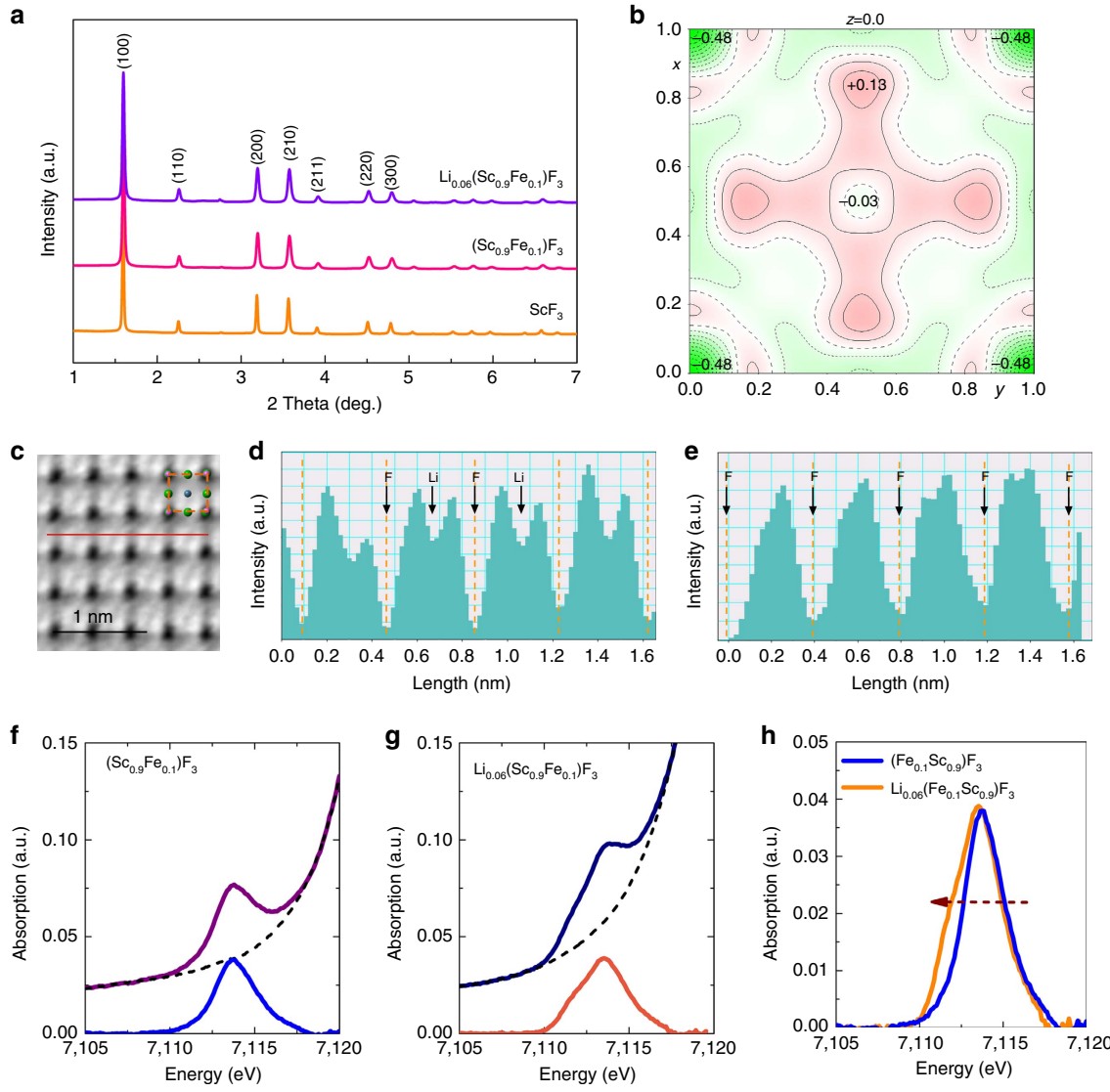

**Figure 2 | The structure and chemical valence of $ScF_3$-based solid solutions.** (**a**) High-energy synchrotron X-ray diffraction patterns of $ScF_3$, $(Sc_{0.9}Fe_{0.1})F_3$ and $Li_{0.06}(Sc_{0.9}Fe_{0.1})F_3$ samples at room temperature. (**b**) Difference Fourier map of $Li_{0.06}(Sc_{0.9}Fe_{0.1})F_3$ at room temperature which was obtained by neutron powder diffraction. The negative intensity indicates that lithium ions are at the A-site. (**c**) ABF image of lithiated region in the $Li_{0.06}(Sc_{0.9}Fe_{0.1})F_3$. The inset shows the arrangements of atoms. (**d**) The corresponding ABF in line profile acquired along the line in **c**. The black arrows mark the fluorine and lithium atomic sites. (**e**) The corresponding ABF in line profile in $(Sc_{0.9}Fe_{0.1})F_3$ without Li as a comparison with **d**. (**f,g**) Fe $K$ pre-edge peak extraction for the $(Sc_{0.9}Fe_{0.1})F_3$ and $Li_{0.06}(Sc_{0.9}Fe_{0.1})F_3$. (**h**) The comparison of Fe $K$ pre-edge peak for both $(Sc_{0.9}Fe_{0.1})F_3$ and $Li_{0.06}(Sc_{0.9}Fe_{0.1})F_3$ after the background subtraction. The ratio $Fe^{3+}/\sum Fe$ can be estimated according to the pre-edge centroid and is reduced in $Li_{0.06}(Sc_{0.9}Fe_{0.1})F_3$ by lithium intercalation.

material provides an effective method for tuning thermal expansion properties, in particular in transforming a NTE precursor to a ZTE or PTE product. Intercalated ions play a critical role by sterically hindering the transverse vibrations of corner-shared $ScF_6$ framework polyhedra, and thus changes overall thermal expansion from negative to positive. This method should be applicable to NTE frameworks containing reducible cations or chemical substitution can be used to introduce such species.

## Methods

**Sample preparation.** The $(Sc_{0.9}Fe_{0.1})F_3$ (SFF) and $ScF_3$ samples were prepared via the solid-state reaction with the precursors of high purity (99.99%) $Sc_2O_3$, $Fe_2O_3$ and $NH_4F$. These precursors in stoichiometric proportions were pressed into a small pellet (radius and height $5 \times 5$ mm) and covered with $NH_4F$ powder and pressed into a larger pellet ($10 \times 10$ mm). The pellet was loaded into Pt crucible and transferred to a furnace with heating at 600 °C for 5 h, and slow cooling to room temperature. The SFF sample was found to be phase pure by X-ray diffraction.

The $Li_{0.06}(Sc_{0.9}Fe_{0.1})F_3$ sample was obtained by the chemical intercalation of the SFF powder with $n$-butyllithium (1.6 M in hexane Aldrich, approximately 10 times the stoichiometric mole ratio) at room temperature for 24 h in a glove box with high purity argon atmosphere. The as-prepared sample powder was washed with hexane several times, and then dried under $N_2$ flow at 80 °C for 10 h. The product did not decompose on heating up to 425 K giving composition $Li_{0.06}(Sc_{0.9}Fe_{0.1})F_3$ (sample LSFF-1). Samples $Li_{0.04}(Sc_{0.94}Fe_{0.06})F_3$ (LSFF-2), $Li_{0.02}(Sc_{0.97}Fe_{0.03})F_3$ (LSFF-3) and $ScF_3$ (LSFF-4) were obtained by partial thermal decomposition in an inert $N_2$ atmosphere, respectively, at 475, 525 and 575 K.

A small amount (0.8 wt%) of impurity phase LiF was observed in LSFF-1. Further impurity phases of iron oxides in LSFF-2 to 4 were generated by high-temperature decomposition as shown in Supplementary Table 3. It needs to note that the presence of impurity phases does not affect the intrinsic thermal expansion properties of the $ReO_3$-type phases.

**X-ray and neutron powder diffraction.** Temperature dependent X-ray diffraction data were collected from 150 to 650 K by using a PANalytical, PW 3040-X-PertPro X-ray diffractometer. The lattice constant was refined using a cubic structural model (space group: $Pm\bar{3}m$). Room temperature

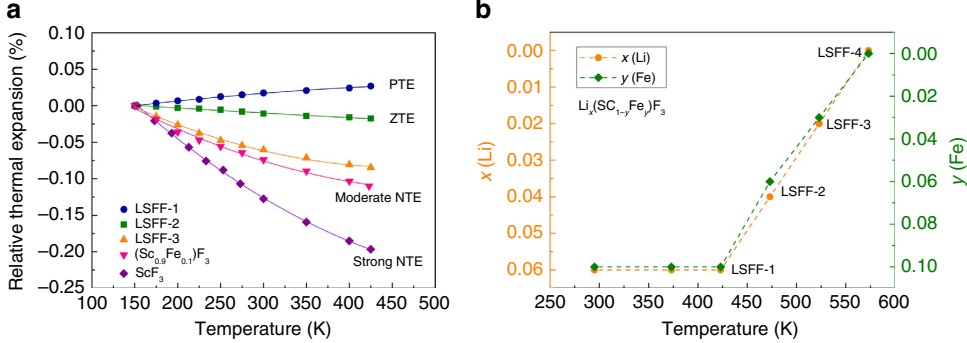

**Figure 3 | The effective control of thermal expansion of $ScF_3$-based compounds.** (**a**) Temperature evolution of relative change in the lattice constant for the PTE LSFF-1 ($Li_{0.06}(Sc_{0.9}Fe_{0.1})F_3$ after annealing 425 K), near ZTE LSFF-2 ($Li_{0.04}(Sc_{0.94}Fe_{0.06})F_3$ after annealing 475 K), moderate NTE LSFF-3 ($Li_{0.02}(Sc_{0.97}Fe_{0.03})F_3$ after annealing 525 K), moderate NTE ($Sc_{0.9}Fe_{0.1})F_3$ and strong NTE $ScF_3$. Errors are smaller than the size of data symbols. (**b**) The stoichiometry of Li and Fe in the main $ReO_3$-type phase, $Li_x(Sc_{1-y}Fe_y)F_3$, for the $Li_{0.06}(Sc_{0.9}Fe_{0.1})F_3$ sample as function of temperature. The stoichiometric values of Li and Fe were estimated according to the content of all phases by means of full-profile Rietveld refinements of neutron powder diffraction (LSFF-1: $Li_{0.06}(Sc_{0.9}Fe_{0.1})F_3$, LSFF-2: $Li_{0.04}(Sc_{0.94}Fe_{0.06})F_3$, LSFF-3: $Li_{0.02}(Sc_{0.97}Fe_{0.03})F_3$, and LSFF-4: $ScF_3$).

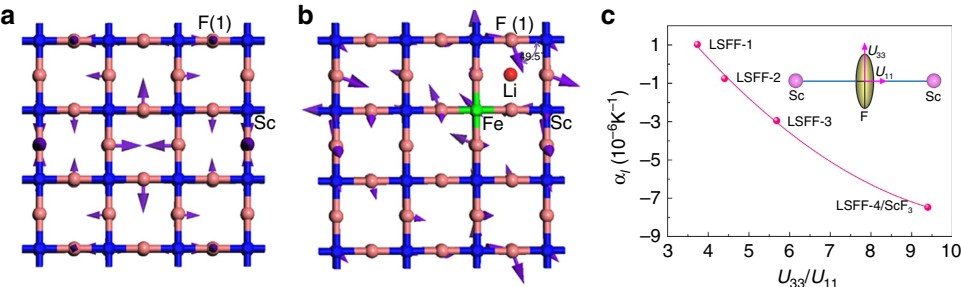

**Figure 4 | The role of Li intercalation in control of thermal expansion of $ScF_3$-based solid solutions.** (**a**) Transverse vibration of fluorine perpendicular to the Sc–F–Sc linkage in the NTE $ScF_3$ super cell. (**b**) Vibration of fluorine deviating from the perpendicular direction to the Sc–F–Sc linkage in the PTE $Li(Sc_{26}Fe)F_{81}$ super cell, corresponding to the composition of $Li_{0.037}(Sc_{0.963}Fe_{0.037})F_3$. Purple arrows indicate the vibration direction of fluorine ions. (**c**) The correlation between CTE and $U_{33}/U_{11}$. $U_{33}$ and $U_{11}$ are atomic displacement parameters of fluorine ions for the transverse and longitudinal directions, respectively. Larger values of $U_{33}/U_{11}$ correspond to greater transverse thermal vibration amplitudes of fluorine ions. The inset shows a schematic thermal ellipsoid of fluorine in the Sc–F–Sc linkage.

synchrotron X-ray diffraction data were collected at the instrument 11-ID-C at the Advanced Photon Source with a wavelength of 0.111650 Å. NPD data of the $Li_{0.06}(Sc_{0.9}Fe_{0.1})F_3$ sample was collected from 298 to 573 K at the NIST Center for Neutron Research on the BT-1 high-resolution neutron powder diffractometer. The wavelength of the neutron beam was 1.5398 Å. The chemical compositions and anisotropic atomic displacement parameters of fluorine ions of the LSFF samples were determined by structure refinement using NPD data. All structure calculations were performed using FULLPROF software[42]. The details of refinements can be found in Supplementary Note 1.

**X-ray absorption fine structure spectroscopy.** Sc and Fe K-edge XANES and extended X-ray absorption fine structure (EXAFS) spectroscopy measurements were performed from room temperature to about 700 K at the XAFS beamline of ELETTRA synchrotron radiation facility in Trieste (Italy). The samples for EXAFS were prepared by mixing and pelletizing the SFF and $Li_{0.06}(Sc_{0.9}Fe_{0.1})F_3$ powder with boron nitride powder. The EXAFS spectra were collected in transmission mode in the energy range 4.3–5.7 keV for Sc K-edge EXAFS, 6.9–8.3 keV for Fe K-edge EXAFS, with an energy step varying from 0.1 eV in the near-edge region to about 4 eV at the highest energies, to obtain a uniform wave vector step $\Delta k \cong 0.03\,Å^{-1}$. The X-ray beam was monochromatized by a Si(111) double-crystal monochromator. The sample was mounted in a high-temperature furnace and the temperature was stabilized and monitored through an electric heater controlled by a feedback loop, ensuring a thermal stability within $\pm 1$ K.

**Scanning transmission electron microscopy.** STEM was performed using a JEM-ARM 200F (JEOL, Tokyo, Japan) that operated at 200 kV and was equipped with double aberration-correctors for probe-forming. Imaging lenses were used to perform high-angle annular-dark field (HAADF) and ABF imaging. The attainable spatial resolution of the microscope is 78 pm at the incident semi-angle of 25 mrad.

To observe Li ions directly using ABF collection geometry, the acceptance semi-angle in this study was fixed between 12 and 25 mrad.

**First-principles calculation.** The first-principles vibrational analysis was performed by CASTEP[43], a total energy package based on the plane-wave pseudopotential density functional theory method[44,45]. The exchange-correlation functional developed by Perdew, Burke and Ernzerhof[46] in general gradient approximation form ref. 47 was adopted to describe the exchange-correlation energy. The effective interaction between the valence electrons (Li $2s^1$, Sc $3d^14s^2$, Fe $3d^64s^2$ and F $2s^22p^5$) and atom cores were modelled by optimized norm-conversing pseudopotentials[48], which allow us to choose a relatively small plane wave basis set without compromising the computational accuracy. The kinetic energy cutoff 900 eV and dense Monkhorst-Pack[49] k-point mesh spanning less than $0.04\,Å^{-1}$ in the Brillouin zone were chosen. To consider the effect of the intercalation of Li ions on the vibrational property, a $3 \times 3 \times 3$ super cell was built in which one Sc atom was replaced by Fe atom and one Li atom was inserted in the neighbouring A-site. Before vibrational property calculation, the crystal structure was geometrically optimized to find the energy minimum. The Broyden–Fletcher–Goldfarb–Shanno (BFGS) minimization scheme[50] was employed in the geometry optimization, in which the convergence criteria for the structure optimization were set to $5.0 \times 10^{-5}$ eV per atom, $0.1$ eV $Å^{-1}$, 0.2 GPa and $5.0 \times 10^{-3}$ GPa for energy, maximum force, maximum stress and maximum displacement, respectively. On the basis of crystal configuration in the minimal energy, the frequencies of the phonon modes were calculated by linear response formalism[51], and the phonon modes were obtained by the second derivative of the total energy with respect to a given perturbation.

**Data availability.** The data relevant to the findings of this study are available from the corresponding authors on reasonable request.

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

## Acknowledgements

This work was supported by the National Natural Science Foundation of China (grant nos. 21322102, 91422301, 21231001, 21590793 and 11474292), the Program for Changjiang Scholars and the Innovative Research Team in University (IRT1207), the Changjiang Young Scholars Award, National Program for Support of Top-notch Young Professionals, and the Fundamental Research Funds for the Central Universities, China (FRF-TP-14-012C1), the Special Foundation of the Director of Technical Institute of Physics and Chemistry (TIPC). The use of the Advanced Photon Source at Argonne National Laboratory was supported by the U.S. Department of Energy, Office of Science, Office of Basic Energy Sciences (DE-AC02-06CH11357). We acknowledge the ELETTRA Synchrotron Radiation Facility for provision of synchrotron radiation as well as all the staff of the XAFS beamline.

## Author contributions

J.C., J.P.A. and X.X. initiated and designed the research. J.C., J.P.A. and X.X. discussed and wrote the manuscript, and all authors revised and commented on it. J.C., Q.G., K.L. and Q.H. measured and analysed the neutron diffraction data. Q.G. synthesized the samples. Q.G. and J.D. measured and analysed the thermal expansion data. A.S., A.C., C.G.R. and L.O. collected the data of EXAFS and A.S. analysed the data of EXAFS. L.G. measured and helped to analyse the STEM data. X.J., Z.L., L.W. and C.W. performed the first-principle calculations, and X.J. wrote the content of theoretical calculations. Y.R. and L.H. measured and helped to analyse the data of synchrotron X-ray diffraction. J.C. and X.X. guided the projects.

**Additional information**

**Competing financial interests:** The authors declare no competing financial interests.

