## [Peer Review File · Nature Communications]

Reviewers' comments:

Reviewer #1 (Remarks to the Author):

This is a very thorough and interesting piece of work, well written and presented. The authors cite appropriate previous studies which show the general concept of control of thermal expansion in framework materials via intercalation/composition change, the novelty here is the process of doing this via controllable redox chemistry rather than, e.g., hydration changes. This will make it a more generally applicable route for metal oxide frameworks. I feel it meets the criteria for Nature Communications and should be published after some minor points are addressed.

1. I would suggest changing the title to make the redox aspect clearer, so perhaps "Tunable thermal... through redox intercalation"?
2. It is incorrect to refer to the work in reference 27 as being on ITQ-29. ITQ-29 is the purely siliceous end-member of the family so would not contain any exchangeable cations and is hydrophobic. Replace "zeolite ITQ-29" with either "zeolite A" or, more correctly, "zeolite LTA"
3. On the third page there is a comparison of lattice parameters before and after intercalation, the numbers need to have esd's stated
4. The recent paper by Senn et al in JACS (<http://dx.doi.org/10.1021/jacs.5b13192>) should be cited

Reviewer #2 (Remarks to the Author):

I believe the authors' experimental reports that changing the chemical composition, both by substitutions on the framework and by insertions of interstitials, diminishes or eliminates the negative thermal expansion of ScF₃. This is of interest for materials engineering.

After substituting some Fe for Sc in the cubic ScF₃ structure, the authors report a significant change in thermal expansion. There is also a large change the thermal expansion after Li is inserted. The samples are unstable above 425 K, so heating above this temperature removes some Fe and Li, reducing the negative thermal expansion. Probably for charge balance, the Li and Fe concentrations are correlated, so changing the Fe changes the Li. The method of composition modification seems successful, although working with n-butyllithium is not for the inexperienced. The authors state that Li insertion and extraction could perhaps be done electrochemically, but it remains to be seen if the material could survive electrochemical cycling.

Overall, the manuscript is well written. However, I have concerns about interpretations, and concerns about novelty.

The computations of phonon dynamics are two levels below what is needed, so this computational work should be removed from the manuscript. In essence, linear response methods cannot give correct results for an anharmonic system like ScF₃. For some time, many other authors have taken the next step to obtain mode Grüneisen parameters by phonon calculations with lattice expansion and contraction, and then use the Grüneisen parameters to discuss anomalous phonon modes. The present authors do not take this typical next step. For highly anharmonic systems, this next step is dubious anyhow. To calculate reliable results for anharmonic phonons, a further step into much more difficult ab initio molecular dynamics calculations is required. (Even many-body phonon theory will be inadequate.) This could be considered beyond the scope of an experimental paper. Removing the computational sections will leave a hole in the discussion, especially since there are no measurements of phonons, but the computational work is just too naive to be

included with their experimental work. It seems that the authors want to support the idea of mechanical bumping and damping of spheres in Fig. 1c, but this model is far from justified by the authors' calculations. ("Damping" as in lines 74 and 86, is not a valid concept for discrete phonons, of course.)

The caption of Supplemental Fig. 9 claims that the number of transverse phonon modes for F atoms is "reduced by 3% after the intercalation of Li ions, indicating that the amplitude of transverse vibration from fluorine is reduced." How were the transverse modes isolated from the overlapping longitudinal modes? (All that is presented are DOS curves.) Even assuming this separation can be made to better than 3% accuracy for off-symmetry phonons with mixed polarizations, 3% is a very small effect. More important would be any change in frequency of the F vibrations with composition or temperature. A conventional next step would be for the authors to integrate the phonon frequency over the green regions in Supplemental Fig. 9, weighted by a Planck factor, to see if there is any difference in average frequency at room temperature. Although a next step, it would be a dubious one, however, since the phonon DOS varies significantly with temperature for anharmonic solids and the authors' calculations were done for 0 K. Especially because the phonons are calculated for 0 K and the interest in them is for temperature effects, my recommendation is to delete Fig. 9 and the figures below.

I presume that Figure 4 was generated from one eigenvector of the dynamical matrix -- i.e., a polarization vector for one phonon. This is only one mode in a vast continuum. In pure cubic ScF₃, the F⁻ ions vibrate in all directions, both parallel and perpendicular to the Sc-F bond. It is true that there is more of a tendency for thermal displacements perpendicular to the bond in ScF₃, but by presenting only one mode for a high symmetry k-point, Fig. 4a and 4b are misleading. Maybe the authors could show thermal ellipsoids from their diffraction results, which might differ between the two materials? Likewise Fig. 1 is an over-reach.

For comparison in Fig. 2d, could the authors please show a calculated image without the presence of Li? In Fig 2e, why is the first doublet so much smaller than the fourth, and why do the shapes of the doublets vary so much?

The underlying idea of controlling thermal expansion by chemical substitution is still somewhat novel, but by now there has been quite a bit of work on altering negative thermal expansion by chemical modification. The authors cite several such papers. They should include at least those in the following list that report tuning of negative thermal expansion of ScF₃ or the ReO₃ structure by chemical modifications:

Morelock C.R., et. al., *J Solid State Chem.* 222, 96 (2015). Shows that framework substitutions have large effects on NTE of ScF₃, so the author's claim on line 63 should be modified. (Their own results also show a large effect for Fe substitution without Li.)

Morelock C.R., et. al., *J Appl. Phys.* 114:213501-1-213501-8 (2013). clear example.

Wilkinson, A.P., *J. Solid State Chem.* 213, 38 (2014). They changed the composition by heating, altering the thermal expansion. (Same structure as ScF₃)

Hancock, J.C., et al. *Chem. Mater.* 27, 3912 (2015). They compared two compositions of the same structure, but perhaps this is not so much tuning.

Romano, C.P., et al., *J. Mater. Sci.* 50, 3409 (2015). They studied the phonon thermodynamics of NTE alloys.

Likely my quick search missed other studies that used chemical modification to alter the thermal expansion of materials with the ReO₃ structure. Also, I did not look for publications on modifying other framework materials such as cyanide structures or MOFs, but I expect more such work has

been done.

I recommend that the authors rewrite this manuscript as an experimental report without speculative phonon dynamics interpretations. The authors did not measure phonons, and their calculations are at least two levels below what is appropriate for highly anharmonic crystals. They also need to do a better job of citing the growing body of work on tuning thermal expansion by chemical modification. The experimental results are interesting. They are a contribution to materials science and engineering, and are somewhat novel.

Reviewer #3 (Remarks to the Author):

This manuscript constitutes an exciting new direction in controlling the thermal expansion of negative thermal expansion materials by redox intercalation of lithium in scandium fluoride. The lithium occupies empty cubic sites and interferes with the transverse vibrations of corner-shared F- atoms. The magnitude of expansion can be tuned between negative, zero and positive values based on the amount of lithium in the structure. The idea is very elegant in its simplicity, yet completely unprecedented to my knowledge. It is certain to be of interest to a wide readership. References are appropriate.

The experimental characterization appears for the most part quite thorough and appropriate, the one thing I am missing is a clear statement on how the lithium content in these materials was determined. It appears as if this was mainly based on neutron diffraction/occupancy refinements? While this method is certainly valid, a separate wet-chemical determination would have been nice. Also, there is no statement about whether Li_{0.06} was the limit of what amount of lithium can be inserted into the material, nor are there enough experimental details given for others to repeat the preparation (e.g., quantities are lacking).

I am a little concerned that the main manuscript gives the impression that single phase samples are investigated, yet when reading the supplemental material, I had to find out that the samples contained an impurity phase. This impurity phase should (i) be mentioned in the main manuscript, and (ii) the quantity should be revealed. What does "tiny" mean? 1%? Less? More? Similarly, a table summarizing the quantities of all phases formed in the annealed samples would be nice to include in SI. In addition (which goes back to my previous comment on whether 0.06 is the limit, how the amount of inserted lithium was controlled or not), it would be beneficial to demonstrate that other compositions (preferably with no or negligible impurity phases) can be prepared directly, whether by chemical or electrochemical intercalation. While the current approach does not negate the idea of tuning intrinsic expansion coefficients through Li content, the more general claim of this being a route to "controlled thermal expansion materials for applications" requires the ability to prepare phase pure materials, as mixtures would display different expansion than what is reported here. A thorough study is obviously not the purpose of this manuscript nor necessary, but the ability to obtain such different composition materials without several impurity phases should at least be addressed.

The manuscript is overall quite readable, and portions are very well written. However, there are also portions (even more so in SI, but also in the main manuscript) that should be corrected by a native English speaker. As at least one co-author is a native English speaker, it is kind of sad that this was not taken care of before submission, but it should definitely be remedied before publication.

A few minor suggestions:

1) I would recommend reporting CTEs in units more commonly found in the literature, e.g., ppm, $\times 10^{-6} \text{ K}^{-1}$ or such - while "per Megakelvin" expresses the same, it is a unit that will require many people to stop and think, and given that "Megakelvin" are not easily reachable, the unit just

does not seem very appealing!

2) On page 8, in Discussion, lines 6-9 are repetitive from the introduction. One occurrence should be removed.

3) On p. 10, "temperature dependence of X-ray diffraction data... were collected" is a very odd way of phrasing things. "Temperature dependent X-ray diffraction data were collected" would be better.

Answers to the reviewer #1:

Comment 1: This is a very thorough and interesting piece of work, well written and presented. The authors cite appropriate previous studies which show the general concept of control of thermal expansion in framework materials via intercalation/composition change, the novelty here is the process of doing this via controllable redox chemistry rather than, e.g., hydration changes. This will make it a more generally applicable route for metal oxide frameworks. I feel it meets the criteria for Nature Communications and should be published after some minor points are addressed.

Answer: The authors appreciate your positive comments on our study.

Comment 2: I would suggest changing the title to make the redox aspect clearer, so perhaps "Tunable thermal... through redox intercalation"?

Answer: We have changed the title to "Tunable thermal expansion in framework materials through redox intercalation" according to your comment, which will be clearer to express the meaning of the manuscript.

Comment 3: It is incorrect to refer to the work in reference 27 as being on ITQ-29. ITQ-29 is the purely siliceous end-member of the family so would not contain any exchangeable cations and is hydrophobic. Replace "zeolite ITQ-29" with either "zeolite A" or, more correctly, "zeolite LTA".

Answer: In the revised manuscript, "zeolite ITQ-29" has been replaced with "zeolite LTA".

Comment 4: On the third page there is a comparison of lattice parameters before

and after intercalation, the numbers need to have esd's stated.

Answer: Lattice parameters were calculated from structure refinements. The lattice parameter changes from 3.99368(5) to 3.99927(4) Å after intercalation, which has been added to the manuscript.

Comment 5: The recent paper by Senn et al in JACS (<http://dx.doi.org/10.1021/jacs.5b13192>) should be cited.

Answer: The JACS paper by Senn et al has reported an effective chemical control of thermal expansion, which has been cited as reference 7 in the manuscript (Senn M. S. *et al.* Symmetry switching of negative thermal expansion by chemical control. *J. Am. Chem. Soc.* **138**, 5479-5482 (2016).)

Answers to the reviewer #2:

Comment 1: I believe the authors' experimental reports that changing the chemical composition, both by substitutions on the framework and by insertions of interstitials, diminishes or eliminates the negative thermal expansion of ScF₃. This is of interest for materials engineering.

Answer: The authors appreciate your positive comments on our study.

Comment 2: After substituting some Fe for Sc in the cubic ScF₃ structure, the authors report a significant change in thermal expansion. There is also a large change the thermal expansion after Li is inserted. The samples are unstable above 425 K, so heating above this temperature removes some Fe and Li, reducing the negative thermal expansion. Probably for charge balance, the Li and Fe concentrations are correlated, so changing the Fe changes the Li. The method of composition modification seems successful, although working with n-butyllithium is not for the inexperienced. The authors state that Li insertion and extraction could perhaps be done electrochemically, but it remains to be seen if the material could survive electrochemical cycling.

Answer: In the present study, we carried out Li intercalation with n-butyllithium

and deintercalation by thermal annealing. We expect Li insertion and extraction could be done by electrochemical method, due to the previous studies on the electrochemical properties of FeF_3 which has a similar ReO_3 -type structure. The reaction mechanisms has been investigated in the literatures.^{1,2} If control the low discharge potential limits to up 2.5 V, corresponding to the reduction of Fe^{3+} to Fe^{2+} via Li insertion reaction, FeF_3 has a good electrochemical cycle performance. The Li insertion and extraction depend on charge-discharge potential. We expect that Li insertion and extraction will be carried out in future studies, and also hope there will be cooperation to study this issue.

[1] Liu, P. *et al.* Thermodynamics and kinetics of the Li/ FeF_3 reaction by electrochemical analysis. *J. Phys. Chem. C*. **116**, 6467-6473 (2012).

[2] Doe, R. E., Persson, K. A., Meng, Y. S. & Ceder, G. First-principles investigation of the Li-Fe-F phase diagram and equilibrium and nonequilibrium conversion reactions of iron fluorides with lithium. *Chem. Mater.* **20**, 5274-5283 (2008).

Comment 3: The computations of phonon dynamics are two levels below what is needed, so this computational work should be removed from the manuscript. In essence, linear response methods cannot give correct results for an anharmonic system like ScF_3 . For some time, many other authors have taken the next step to obtain mode Grueneisen parameters by phonon calculations with lattice expansion and contraction, and then use the Grueneisen parameters to discuss anomalous phonon modes. The present authors do not take this typical next step. For highly anharmonic systems, this next step is dubious anyhow. To calculate reliable results for anharmonic phonons, a further step into much more difficult ab initio molecular dynamics calculations is required. (Even many-body phonon theory will be inadequate.) This could be considered beyond the scope of an experimental paper. Removing the computational sections will leave a hole in the discussion, especially since there are no measurements of phonons, but the computational work is just too naive to be included with their experimental work. It seems that the authors want to support the idea of

mechanical bumping and damping of spheres in Fig. 1c, but this model is far from justified by the authors' calculations. ("Damping" as in lines 74 and 86, is not a valid concept for discrete phonons, of course.)

The caption of Supplemental Fig. 9 claims that the number of transverse phonon modes for F atoms is "reduced by 3% after the intercalation of Li ions, indicating that the amplitude of transverse vibration from fluorine is reduced." How were the transverse modes isolated from the overlapping longitudinal modes? (All that is presented are DOS curves.) Even assuming this separation can be made to better than 3% accuracy for off-symmetry phonons with mixed polarizations, 3% is a very small effect. More important would be any change in frequency of the F vibrations with composition or temperature. A conventional next step would be for the authors to integrate the phonon frequency over the green regions in Supplemental Fig. 9, weighted by a Planck factor, to see if there is any difference in average frequency at room temperature. Although a next step, it would be a dubious one, however, since the phonon DOS varies significantly with temperature for anharmonic solids and the authors' calculations were done for 0 K. Especially because the phonons are calculated for 0 K and the interest in them is for temperature effects, my recommendation is to delete Fig. 9 and the figures below.

I presume that Figure 4 was generated from one eigenvector of the dynamical matrix -- i.e., a polarization vector for one phonon. This is only one mode in a vast continuum. In pure cubic ScF_3 , the F ions vibrate in all directions, both parallel and perpendicular to the Sc-F bond. It is true that there is more of a tendency for thermal displacements perpendicular to the bond in ScF_3 , but by presenting only one mode for a high symmetry k-point, Fig. 4a and 4b are misleading. Maybe the authors could show thermal ellipsoids from their diffraction results, which might differ between the two materials? Likewise Fig. 1 is an over-reach.

Answer: We comply with these comments to revise the part of the computations of phonon dynamics. The original Supplemental Fig. 9 and related discussions have been removed. We keep Fig. 4 of the main text, and have added thermal ellipsoids of F ions to explain the mechanism of intercalated Li ions to thermal expansion.

We have calculated all of the phonon modes and performed comparison before and after Li intercalation. It is observed that almost all the transverse vibrations of F ions in pure ScF₃ are inclined after the Li intercalation. However, due to the large number of transverse vibrational modes, it is impractical to display evolution of all the modes, so we show a representative one, that with the lowest frequency to schematically describe the effect of Li intercalation on the vibration of F ions.

We have added the following content in the section for the first-principles calculation: “The results reveal that the vibrations of fluorine ions are strongly perturbed by the inserted Li ions, and the directions of all transverse modes are inclined as shown the representative mode with the lowest frequency in Fig. 4a and b.” In the revised manuscript, we keep the original Fig. 4, since this picture can give a useful understanding to the reader of the effect of Li intercalation on thermal expansion.

The advice one thermal ellipsoids of F ions is useful to help explain for the effect of Li intercalation on thermal expansion. We have refined the anisotropic atomic displacement parameters of F ions for the samples of LSFF-1, LSFF-2, LSFF-3, and LSFF-4 (ScF₃) using neutron powder diffraction (NPD) data. Supplementary Table 4 lists the values of anisotropic atomic displacement parameters of F ions and CTEs of the LSFF compositions. With increasing content of Li the transverse thermal vibration of F ions (U_{33}) is weakened, while the longitudinal one (U_{11}) is enhanced. This means that Li intercalation hinders the transverse vibration of F ions. As shown in Fig. 4c, there is a good correlation between CTE and the ratio of U_{33}/U_{11} . Larger values of U_{33}/U_{11} indicate stronger transverse thermal vibrations of fluorine ions, and with increasing U_{33}/U_{11} , the thermal expansion changes from positive to negative.

Comment 4: For comparison in Fig. 2d, could the authors please show a calculated image without the presence of Li? In Fig. 2e, why is the first doublet so much smaller than the fourth, and why do the shapes of the doublets vary so much?

Answer: We have compared the ABF images of samples with and without Li ions as now shown in Figs. 2d and 2e. We have carried out the experiment for the ABF

image of $(\text{Sc}_{0.9}\text{Fe}_{0.1})\text{F}_3$ without Li ions (Fig. 2e). If we compare the ABF in line profiles shown in Fig. 2d and Fig. 2e, we can see that Li ions have been inserted into the lattice of $(\text{Sc}_{0.9}\text{Fe}_{0.1})\text{F}_3$.

For the second question for the original Fig. 2e, we think the most possible reason could be due to the facts that Li ions were randomly inserted into the lattice of $(\text{Sc}_{0.9}\text{Fe}_{0.1})\text{F}_3$, and the crystal structure of this powder material is not perfect. The results of ABF image would like to give the evidence for the presence of intercalated Li ions, but it is difficult to give accurate quantitative information from the intensity. The asymmetry in some of the observed “doublet” is often observed for ABF line profiles, as observed in the literatures below¹⁻³.

[1] Wang, R. *et al.* Atomic structure of Li_2MnO_3 after partial delithiation and re-lithiation. *Adv. Energy Mater.* **3**, 1358–1367 (2013).

[2] Gu, L. *et al.* Direct observation of lithium staging in partially delithiated LiFePO_4 at atomic resolution. *J. Am. Chem. Soc.* **133**, 4661–4663 (2011).

[3] Yue, J.-L. *et al.* Discrete Li-occupation versus pseudo-continuous Na-occupation and their relationship with structural change behaviors in $\text{Fe}_2(\text{MoO}_4)_3$. *Sci. Rep.* **5**, 8810 (2015).

Comment 5: The underlying idea of controlling thermal expansion by chemical substitution is still somewhat novel, but by now there has been quite a bit of work on altering negative thermal expansion by chemical modification. The authors cite several such papers. They should include at least those in the following list that report tuning of negative thermal expansion of ScF_3 or the ReO_3 structure by chemical modifications:

Morelock C. R., et al., *J Solid State Chem.* 222, 96 (2015). Shows that framework substitutions have large effects on NTE of ScF_3 , so the author's claim on line 63 should be modified. (Their own results also show a large effect for Fe substitution without Li.)

Morelock C. R., et. al., *J Appl. Phys.* 114:213501-1–213501-8 (2013). clear example.

Wilkinson, A. P., J. Solid State Chem. 213, 38 (2014). They changed the composition by heating, altering the thermal expansion. (Same structure as ScF₃)

Hancock, J. C., et al. Chem. Mater. 27, 3912 (2015). They compared two compositions of the same structure, but perhaps this is not so much tuning.

Romano, C. P., et al., J. Mater. Sci. 50, 3409 (2015). They studied the phonon thermodynamics of NTE alloys.

Likely my quick search missed other studies that used chemical modification to alter the thermal expansion of materials with the ReO₃ structure. Also, I did not look for publications on modifying other framework materials such as cyanide structures or MOFs, but I expect more such work has been done.

Answer: We have revised the manuscript according to these comments, and the papers have been cited, also some others which reported control of thermal expansion of ReO₃ type materials and other framework materials such as cyanides and MOFs. However, we note that all of these papers describe substitutions within the frameworks, rather than by the intercalation method we report here.

Comment 6: I recommend that the authors rewrite this manuscript as an experimental report without speculative phonon dynamics interpretations. The authors did not measure phonons, and their calculations are at least two levels below what is appropriate for highly anharmonic crystals. They also need to do a better job of citing the growing body of work on tuning thermal expansion by chemical modification. The experimental results are interesting. They are a contribution to materials science and engineering, and are somewhat novel.

Answer: We hope that the above rewrites have addressed these issues.

Answers to the reviewer #3:

Comment 1: This manuscript constitutes an exciting new direction in controlling the thermal expansion of negative thermal expansion materials by redox intercalation of lithium in scandium fluoride. The lithium occupies empty cubic sites and interferes with the transverse vibrations of corner-shared F-atoms. The magnitude of expansion

can be tuned between negative, zero and positive values based on the amount of lithium in the structure. The idea is very elegant in its simplicity, yet completely unprecedented to my knowledge. It is certain to be of interest to a wide readership. References are appropriate.

Answer: The authors appreciate your positive comments on our study.

Comment 2: The experimental characterization appears for the most part quite thorough and appropriate. The one thing I am missing is a clear statement on how the lithium content in these materials was determined. It appears as if this was mainly based on neutron diffraction/occupancy refinements? While this method is certainly valid, a separate wet-chemical determination would have been nice. Also, there is no statement about whether $\text{Li}_{0.06}$ was the limit of what amount of lithium can be inserted into the material, nor are there enough experimental details given for others to repeat the preparation (e.g., quantities are lacking).

Answer: The original compositions were determined from occupancy refinements using NPD data. For the revised manuscript, we have carried out additional inductively coupled plasma spectroscopy (ICP) analysis. For LSFF-1, the measured atom ratio of Sc : Fe : Li is 1.0 : 0.11 : 0.12. After deducting the contribution from the impurity phase of LiF, the chemical composition is $\text{Li}_{0.065}(\text{Sc}_{0.9}\text{Fe}_{0.099})\text{F}_3$ with normalization of Sc stoichiometry to 0.9. This is in good agreement with the results from NPD data.

In the present experiment $\text{Li}_{0.06}$ should be the limit of the amount of Li ions which can be inserted into the material of $(\text{Sc}_{0.9}\text{Fe}_{0.1})\text{F}_3$, because a plenty of n-butyllithium (approximately 10 times) was used to reduce the sample of $(\text{Sc}_{0.9}\text{Fe}_{0.1})\text{F}_3$. Furthermore, the reduction process was taken by liquid-solid reaction and for a sufficient time (24 h). The sample of $(\text{Sc}_{0.9}\text{Fe}_{0.1})\text{F}_3$ could be inserted with Li ions with a saturated concentration.

In the revised manuscript, the experimental details are shown in the part of sample preparation of the Methods section.

Comment 3: I am a little concerned that the main manuscript gives the impression that single phase samples are investigated, yet when reading the supplemental material, I had to find out that the samples contained an impurity phase. This impurity phase should (i) be mentioned in the main manuscript, and (ii) the quantity should be revealed. What does "tiny" mean? 1%? Less? More? Similarly, a table summarizing the quantities of all phases formed in the annealed samples would be nice to include in SI.

Answer: The phase content was determined by NPD refinement, as listed in Supplementary Table 3. The corresponding revisions have been made in the main text and supplementary information, and phase content (wt%) of is also stated.

Comment 4: In addition (which goes back to my previous comment on whether 0.06 is the limit, how the amount of inserted lithium was controlled or not), it would be beneficial to demonstrate that other compositions (preferably with no or negligible impurity phases) can be prepared directly, whether by chemical or electrochemical intercalation. While the current approach does not negate the idea of tuning intrinsic expansion coefficients through Li content, the more general claim of this being a route to "controlled thermal expansion materials for applications" requires the ability to prepare phase pure materials, as mixtures would display different expansion than what is reported here. A thorough study is obviously not the purpose of this manuscript nor necessary, but the ability to obtain such different composition materials without several impurity phases should at least be addressed.

Answer: The authors appreciate your positive comments on the present idea to tune thermal expansion via ions intercalation. It is important to control the content of inserted Li ions in pure phase. As these comments have pointed out, mixtures would display different thermal expansion. Using the present sample preparation method, the amount of inserted Li ions can be basically controlled by the content of Fe, since only Fe element can be reduced with n-butyllithium. For example if we use the sample of $(\text{Sc}_{0.95}\text{Fe}_{0.05})\text{F}_3$, less amount of Li ions can be inserted which should be lower than the stoichiometry of Fe (5%). In the present manuscript, the amount of inserted Li ions

was controlled by various annealing temperature.

In the revised manuscript, we have addressed the possible methods to obtain samples with pure phase. Samples could be prepared by the solid-state reaction or electrochemical intercalation. For the solid-state reaction, it needs to carefully control the valence of Fe^{2+} and react in inert or reductive atmosphere. The electrochemical intercalation is also a possible way. It needs to understand its electrochemical performance. The amount of inserted Li ions could be controlled according to the charge and discharge curves.

We have rewritten the text to show that even aggressive lithiation via n-butyllithium of the NTE precursors SFF (which is phase pure) give the PTE product LSFF-1 with a trace of LiF (0.8 wt%). In this study we have used thermal decomposition which inevitably creates impurities to study deintercalation, but we expect that this could be done chemically or electrochemically in real applications.

Comment 5: The manuscript is overall quite readable, and portions are very well written. However, there are also portions (even more so in SI, but also in the main manuscript) that should be corrected by a native English speaker. As at least one co-author is a native English speaker, it is kind of sad that this was not taken care of before submission, but it should definitely be remedied before publication.

A few minor suggestions:

1) I would recommend reporting CTEs in units more commonly found in the literature, e.g., ppm, $\times 10^{-6} \text{ K}^{-1}$ or such - while "per Megakelvin" expresses the same, it is a unit that will require many people to stop and think, and given that "Megakelvin" are not easily reachable, the unit just does not seem very appealing!

2) On page 8, in Discussion, lines 6-9 are repetitive from the introduction. One occurrence should be removed.

3) On p. 10, "temperature dependence of X-ray diffraction data... were collected" is a very odd way of phrasing things. "Temperature dependent X-ray diffraction data were collected" would be better.

Answer: The repetitive discussions on page 8 have been removed. The others have

also been corrected according to the comments. The English has been corrected throughout.

REVIEWERS' COMMENTS:

Reviewer #2 (Remarks to the Author):

The authors have improved the manuscript in a number of important ways. I now have few concerns about interpretations, although I still have concerns about novelty.

The authors have de-emphasized the importance of their computations of harmonic phonons. They now present the negative thermal expansion more in the context of previous work, which involves the "guitar string" model where transverse displacements pull on the ends of the string and cause shrinkage. This has been used for some time for ScF_3 , and can probably be accepted by most readers. The authors present new evidence that composition changes alter this behavior by showing the thermal ellipsoids of F atom vibration in Fig. 4c. This is an important new part of the manuscript, and the trend looks convincing. I am happy they took this suggestion from the previous review, and that it worked so well.

(As a minor point, I still object to the statement in the summary that spectral weight is moved from transverse to longitudinal modes. I think what the authors need to say is that the longitudinal and transverse stiffnesses of F-ion bonds are altered by the chemical changes. They do not present enough work on polarization analysis to make a conclusion about the nature of the phonons, however.)

Bringing up the possibility of electrochemical control of Li concentration is appropriate, but it is not obvious if this will work as well as for FeF_3 , or even if it will work at all.

1.) I suspect the authors' electronic structure calculations have shown their materials to be insulators. Materials with insulating ground states such as FeF_3 can conduct electricity (and hence be useful as electrodes) by polaronic mechanisms at finite temperatures. Predicting these temperatures is not practical in 2016. Nevertheless, we know that polaron hopping is suppressed with increasing distance between the multivalent atoms. Unfortunately, the Fe^{2+} and Fe^{3+} in their materials are much further apart than the Fe ions in FeF_3 , so I doubt that polaron hopping is possible in these materials if the valence change is at Fe ions.

2.) The formation of LiF is a real risk, and LiF is electrochemically inactive. This is an issue in cycling FeF_3 , although it is worse as the cells are driven to deeper cycles.

Perhaps, though, it is not necessary for the material to have a NTE that is electrochemically reversible, and the authors' method with n-butyllithium may suffice for future impact.

I return to my concern about novelty. There are so many reports already of altered thermal expansion by chemical modification of ScF_3 , and many other reports of altered thermal expansion by intercalating molecules into crystals with open structures. The authors claim they are first to alter the thermal expansion of ScF_3 by intercalation. This may be true, but even if so, are the results really novel, compared to the known state of the art? Given the state of the art, to me these results seem incremental but fashionable.

Reviewer #3 (Remarks to the Author):

The authors have adequately addressed all my previous comments. The manuscript can be published as is.

Answers to the reviewer #2:

Comment 1: The authors have improved the manuscript in a number of important ways. I now have few concerns about interpretations, although I still have concerns about novelty.

The authors have de-emphasized the importance of their computations of harmonic phonons. They now present the negative thermal expansion more in the context of previous work, which involves the "guitar string" model where transverse displacements pull on the ends of the string and cause shrinkage. This has been used for some time for ScF_3 , and can probably be accepted by most readers. The authors present new evidence that composition changes alter this behavior by showing the thermal ellipsoids of F atom vibration in Fig. 4c. This is an important new part of the manuscript, and the trend looks convincing. I am happy they took this suggestion from the previous review, and that it worked so well.

(As a minor point, I still object to the statement in the summary that spectral weight is moved from transverse to longitudinal modes. I think what the authors need to say is that the longitudinal and transverse stiffnesses of F-ion bonds are altered by the chemical changes. They do not present enough work on polarization analysis to make a conclusion about the nature of the phonons, however.)

Answer: The authors appreciate the positive comments on the explanation for the change of thermal expansion by the experiment data. We have de-emphasized the importance of calculations of harmonic phonons, since we do not present enough work on polarization analysis. We have removed the statement in the summary that spectral weight is moved from transverse to longitudinal modes.

Comment 2: Bringing up the possibility of electrochemical control of Li concentration is appropriate, but it is not obvious if this will work as well as for FeF_3 , or even if it will work at all.

1.) I suspect the authors' electronic structure calculations have shown their materials to be insulators. Materials with insulating ground states such as FeF_3 can

conduct electricity (and hence be useful as electrodes) by polaronic mechanisms at finite temperatures. Predicting these temperatures is not practical in 2016. Nevertheless, we know that polaron hopping is suppressed with increasing distance between the multivalent atoms. Unfortunately, the Fe^{2+} and Fe^{3+} in their materials are much further apart than the Fe ions in FeF_3 , so I doubt that polaron hopping is possible in these materials if the valence change is at Fe ions.

2.) The formation of LiF is a real risk, and LiF is electrochemically inactive. This is an issue in cycling FeF_3 , although it is worse as the cells are driven to deeper cycles. Perhaps, though, it is not necessary for the material to have a NTE that is electrochemically reversible, and the authors' method with n-butyllithium may suffice for future impact.

Answer: The conductivity is an important property for the intercalation of electrode materials. We have conducted two kinds of experiments to answer your questions. Firstly, UV-vis diffuse absorption spectra of $(\text{Sc}_{1-x}\text{Fe}_x)\text{F}_3$ ($x = 0, 0.05, \text{ and } 0.1$) have been measured. ScF_3 has a large band gap ($E_g = 8\sim 10$ eV) which was ever calculated in the previous literature (Zhgun, P. *et al.* arXivpreprint arXiv:1211.5697, (2012)). However, $(\text{Sc}_{0.9}\text{Fe}_{0.1})\text{F}_3$ exhibits a noticeable visible-light absorption (Fig. R1). The E_g of $(\text{Sc}_{0.9}\text{Fe}_{0.1})\text{F}_3$ is estimated to be 1.87 eV, which means that it could have a semiconducting property.

Fig. R1 | UV-vis diffuse absorption spectra of $(\text{Sc}_{1-x}\text{Fe}_x)\text{F}_3$ ($x = 0, 0.05, \text{ and } 0.1$).

The inset shows the color change of the samples.

Secondly, we have tried a preliminary experiment to insert Li ions into $(\text{Sc}_{0.9}\text{Fe}_{0.1})\text{F}_3$ compound by the electrochemical method of Li ion battery technology. After discharging, the material was taken out. The lithiated sample of $\text{Li}_x(\text{Sc}_{0.9}\text{Fe}_{0.1})\text{F}_3$ (LSFF-electrochemical) was washed by a DMC solvent to remove the electrolyte, and then dried in a vacuum chamber connected with glove box. The phase structure and thermal expansion of the lithiated sample were determined by the XRD measurement. We can see that the LSFF-electrochemical sample is stable during the electrochemical method (Fig. R2a). We do observe that the LSFF-electrochemical sample exhibits a weakened NTE with a CTE of $-0.92 \times 10^{-6} \text{ K}^{-1}$ (Fig. R2c) when compared with that of $(\text{Sc}_{0.9}\text{Fe}_{0.1})\text{F}_3$ ($-5.01 \times 10^{-6} \text{ K}^{-1}$). It means Li ions can be inserted in the lattice of $(\text{Sc}_{0.9}\text{Fe}_{0.1})\text{F}_3$ by the electrochemical method.

We may mention that the detailed experiments on Li-intercalation via electrochemical method are beyond the present study scope. We hope that the present study would invite joint studies by researchers from both the fields of electrochemistry and NTE in near future.

Fig. R2 | Structure and thermal expansion of the LSFF-electrochemical sample with inserted Li ions by electrochemical method. (a) Temperature dependence of XRD patterns, and (b) relative thermal expansion of LSFF-electrochemical, $(\text{Sc}_{0.9}\text{Fe}_{0.1})\text{F}_3$, and ScF_3 . Errors are smaller than the size of data symbols.

Comment 3: I return to my concern about novelty. There are so many reports

already of altered thermal expansion by chemical modification of ScF_3 , and many other reports of altered thermal expansion by intercalating molecules into crystals with open structures. The authors claim they are first to alter the thermal expansion of ScF_3 by intercalation. This may be true, but even if so, are the results really novel, compared to the known state of the art? Given the state of the art, to me these results seem incremental but fashionable.

Answer: The effective control of thermal expansion is important issue for the study of NTE. It has well known that the common used method is chemical substitution, not only for the present studied ScF_3 but also other NTE compounds. It is true that there have important progresses for the control of thermal expansion of ScF_3 via chemical substitution. The present study emphasizes one effective way of Li intercalation to control thermal expansion of ScF_3 . This method would be useful for other framework structure NTE compounds, since inserted ions or molecules would have a strong effect on the transverse vibration of linkages or polyhedra. We hope the present work will trigger the study on control of thermal expansion via Li or other ions intercalation for open framework NTE materials in future.